# Controlling the Redox Catalytic Activity of a Cyclic Selenide Fused to 18-Crown-6 by the Conformational Transition Induced by Coordination to an Alkali Metal Ion

**DOI:** 10.3390/molecules28083607

**Published:** 2023-04-20

**Authors:** Michio Iwaoka, Hajime Oba, Takeru Ito

**Affiliations:** 1Department of Chemistry, School of Science, Tokai University, Hiratsuka-shi 259-1292, Kanagawa, Japan; 7bsc1112@gmail.com (H.O.); takeito@keyaki.cc.u-tokai.ac.jp (T.I.); 2Institute of Advanced Biosciences, Tokai University, Hiratsuka-shi 259-1292, Kanagawa, Japan

**Keywords:** selenide, crown ethers, coordination chemistry, conformational analysis, NMR titration, redox catalysis, enzyme mimics, ion sensors

## Abstract

*trans*-3,4-Dihydroxyselenolane (DHS), a water-soluble cyclic selenide, exhibits selenoenzyme-like unique redox activities through reversible oxidation to the corresponding selenoxide. Previously, we demonstrated that DHS can be applied as an antioxidant against lipid peroxidation and a radioprotector by means of adequate modifications of the two hydroxy (OH) groups. Herein, we synthesized new DHS derivatives with a crown-ether ring fused to the OH groups (DHS-crown-*n* (*n* = 4 to 7), **1**–**4**) and investigated their behaviors of complex formation with various alkali metal salts. According to the X-ray structure analysis, it was found that the two oxygen atoms of DHS change the directions from diaxial to diequatorial by complexation. The similar conformational transition was also observed in solution NMR experiments. The ^1^H NMR titration in CD_3_OD further confirmed that DHS-crown-6 (**3**) forms stable 1:1 complexes with KI, RbCl and CsCl, while it forms a 2:1 complex with KBPh_4_. The results suggested that the 1:1 complex (**3**·MX) exchanges the metal ion with metal-free **3** through the formation of the 2:1 complex. The redox catalytic activity of **3** was evaluated using a selenoenzyme model reaction between H_2_O_2_ and dithiothreitol. The activity was significantly reduced in the presence of KCl due to the complex formation. Thus, the redox catalytic activity of DHS could be controlled by the conformational transition induced by coordination to an alkali metal ion.

## 1. Introduction

Crown ethers have been widely utilized as synthetic modules for various functional molecules because they can selectively accommodate a metal ion in the cavity or a polymer chain through the hole [1,2]. In particular, the molecular recognition functions of crown ethers are essential in designing rotaxane-based molecular machines [3,4,5] and molecular sensors [6,7,8,9]. Since the discovery, crown ethers and their analogues that involve chalcogen atoms, i.e., S, Se and Te, instead of the oxygen atoms [1,10,11] have been elaborately studied from various points of view. For example, thia-crown ethers have been employed as extractants for heavy metal ions [12,13,14,15]. Redox responses as well as catalytic functions of some thia- and selena-crown ethers were investigated [16,17,18,19,20,21]. Molecular assemblies through chalcogen bonding interactions have also been studied recently for telluro-ether heterocycles [22].

*trans*-3,4-Dihydroxyselenolane (DHS) is a water-soluble five-membered ring selenide with a selenoenzyme-like unique redox catalytic activity through the reversible oxidation to the selenoxide >Se=O (DHS^ox^) [23,24]. It is also a stable molecule with low toxicity [25,26]. These features are potential merits of DHS when it is applied to medicinal and material sciences. We previously demonstrated that DHS can be an antioxidant against lipid peroxidation [27,28] and a radioprotector [29] by means of adequate chemical modifications of the two hydroxy (OH) groups. In this study, we developed new DHS derivatives with a crown-ether ring fused to the OH groups (DHS-crown-*n* (*n* = 4 to 7), **1**–**4**) and investigated their behaviors as host molecules to various alkali metal salts. Moreover, the redox catalytic activity of one of the new DHS derivatives, i.e., DHS-crown-6 (**3**), was evaluated in a selenoenzyme model reaction. It was found that the activity can be controlled by the complexation with a potassium ion (K^+^).

## 2. Results and Discussion

DHS-crowns (**1**–**4**) were synthesized in the yields of 40 to 59% by the coupling reaction between DHS and the corresponding ditosylates in the presence of NaOH or KOH as a template (Figure 1). The obtained compounds were characterized by ^1^H, ^13^C and ^77^Se NMR as well as MS spectrometry.

The molecular structure of DHS-crown-4 (**1**) was determined by X-ray analysis (Figure 1). The two oxygen atoms of DHS fused by the 12-crown-4 ring occupied axial positions of the five-membered ring of DHS. Similar diaxial conformations are commonly observed for the parent DHS [30] and its derivatives [31,32]. This is probably due to the electrostatic repulsion between the two negatively charged oxygen atoms.

We subsequently investigated the complex formation of **1**–**4** with an alkali metal ion. Fine single crystals were obtained when the host molecules **1**–**3** were mixed in methanol with KI or NaBr as a salt. Figure 2 shows the molecular structures for **1**·0.5KI·H_2_O (**5**), **1**·0.5NaBr·2H_2_O (**6**), **2**·0.5KI (**7**) and **3**·KI (**8**).

For DHS-crown-4 (**1**), 2:1 complexes (**5** and **6**) with KI and NaBr, respectively, were crystallized. Complex **5** showed a distorted sandwich-type shape owing to the smaller cavity of 12-crown-4, which pushed the larger alkali ion (K^+^) out of the cavity. The position of the counter anion (I^–^) was close to K^+^ (*r*_K···I_ = 3.537(1) Å). This weak coordination interaction enforced the two crown-ether rings open in one side. On the other hand, in complex **6**, the counter anion (Br^–^) did not have an interaction with Na^+^, allowing for the two crown-ether rings in an almost parallel configuration. This would be because Na^+^ can enter the cavity of the 12-crown-4 ring deeper than K^+^ due to the smaller ion radius.

A similar parallel sandwich-type structure to complex **6** was observed for complex **7**, in which K^+^ entered the cavity of the 15-crown-5 ring of **2** a little more deeply than in the case of complex **5**. On the other hand, in complex **8**, K^+^ resided in the center of the 18-crown-6 ring of **3**, forming a flat 1:1 complex. In this complex, the counter anion (I^–^) existed close to K^+^ (*r*_K···I_ = 3.413(2) Å), blocking one apical site of K^+^, and the other apical site was occupied by the Se atom of the other DHS-crown-6 molecule (**3**), forming a weak Se···K^+^ interaction (*r*_K···Se_ = 3.398(2) Å). By comparing the structure of **8** with those of complexes **5** and **7**, it is seen that K^+^ was captured by the crown-ether ring more strongly with an increasing ring size.

It is notable that, in all complexes (**5**–**8**), the two oxygen atoms of DHS occupied the equatorial directions. This contrasts with the observation that the oxygen atoms occupied the axial directions in the metal-free form of DHS-crown-4 (**1**) (Figure 1). The conformational transition can be reasonably explained by considering that the two oxygen atoms in the axial positions are too far apart from each other to coordinate to a metal ion. Thus, the DHS ring of DHS-crowns would change the conformation from diaxial to diequatorial by complex formation with an alkali metal ion.

Subsequently, the complexation process of DHS-crown-6 (**3**) was investigated in more detail by NMR to confirm the axial to equatorial conformational transition observed in the solid state. The ^1^H NMR spectrum for **3** in CD_3_OD (Figure 3a) showed the signal for the H_a_ proton of the DHS ring at 4.29 ppm as multiplet, while those for the H_b_ and H_c_ protons at 3.06 and 2.90 ppm were shown as doublet of doublets. These spectral features were almost identical with those observed in the ^1^H NMR spectra for the other DHS-crowns, i.e., **1**, **2** and **4**, suggesting that the metal-free forms of **1**–**4** maintain the same conformation in the solution. However, when KI (1 eq) was added, significant changes in the NMR spectrum were observed (Figure 3b). The signal for H_a_ shifted upfield from 4.29 to 3.95 ppm, and the signals for the CH_2_ protons appeared as largely separated peaks at 3.21 and 2.68 ppm. The profiles of the peaks also became different from those observed before the addition of KI. The spectral changes strongly supported the conformational transition induced by coordination to K^+^. Similarly, significant spectral changes were observed in the ^13^C and ^77^Se NMR spectra for **3** after the addition of KI (see SI).

Figure 4 shows the changes in the peak separation between H_b_ and H_c_ protons of **3** (Δ(H_b_ − H_c_)) in the ^1^H NMR spectrum when a variable amount of an alkali metal salt (MX) was added. The separation width gradually increased with the addition of MX, indicating that the metal ion (M^+^) was rapidly exchanged among the host molecules within a time scale of the NMR measurement. The observed titration curves clearly show that stable 1:1 complexes were completely produced not only with KI but also with CsCl and RbCl when 1 eq of the metal salts were added, because Δ(H_b_ − H_c_) reached plateaus. On the other hand, NaCl seemed to form the 1:1 complex with **3** in equilibrium with the metal-free host, as Δ(H_b_ − H_c_) was still increasing when 2 eqs of the salt were added. The formation of the less stable complex would be due to the smaller size of Na^+^ compared to the 18-crown-6 cavity of **3**. Interestingly, when potassium tetraphenylborate (KBPh_4_) was employed as a salt, the peak separation became a maximum with the addition of 0.5 eq, suggesting the formation of a stable 2:1 complex. This complex would have a sandwich-type structure like those observed for complexes **6** and **7** (Figure 2) and should have become stable because the counter anion (BPh_4_^–^) does not possess a coordinating ability. According to these observations, we propose the mechanism for the rapid exchange of an alkali metal ion between DHS-crown-6 molecules (**3**), as illustrated in Figure 5. In this model, the cation exchange is mediated by the formation of a 2:1 complex, which would be stable when a counter anion (X^−^) is non-coordinating, such as BPh_4_^–^, while it would decompose to the 1:1 complex (**3**·MX) and the metal-free host (**3**) when X^−^ is a coordinating ligand, such as I^–^. Indeed, Δ(H_b_ − H_c_) became saturated before the addition of 1 eq of KI (Figure 4), suggesting the transient formation of the 2:1 complex during the NMR titration.

According to the X-ray and ^1^H NMR studies on the complexation process of DHS-crown-6 (**3**), it is now obvious that the conformation of **3** can be controlled by the coordination to an alkali metal ion. We next planned to apply this interesting property to a molecule device, in which the redox catalytic activity of DHS can be turned on or off by the addition of an alkali metal ion. It is well established that DHS can be oxidized to the selenoxide >Se=O (DHS^ox^) with H_2_O_2_ and DHS^ox^ can be easily reduced back to DHS with thiol (RSH) [24]. This reaction cycle has been used as a model enzyme reaction of selenium-dependent glutathione peroxidase (GPx), because H_2_O_2_, a representative reactive oxygen species, is catalytically degraded to H_2_O. Therefore, the similar redox behaviors were expected for DHS-crowns.

Indeed, host molecule **3** exhibited the GPx-like catalytic activity in the reaction between H_2_O_2_ and dithiothreitol (DTT^red^) (Figure 6, Cycle A). When the reaction was tracked by ^1^H NMR in CD_3_OD at room temperature, the disappearance of DTT^red^ and the formation of the oxidized disulfide (DTT^ox^) were observed with the reaction progression (see SI). The residual amounts of DTT^red^ as a function of the reaction time are graphically shown in Figure 7. The catalytic activity of **3** was almost equal to that of the parent DHS [24]. On the other hand, when KCl was added to **3**, the activity was significantly reduced. The half-life time of DTT^red^ (*t*_1/2_) increased by about fourfold in the presence of KCl (*t*_1/2_ ~240 min vs. ~60 min in the absence of KCl). It should be noted that the presence of KCl did not affect the catalytic activity for the DHS and blank, while KI itself exerted a substantial catalytic activity in this reaction system. The reduction in the catalytic activity of **3** would be due to the conformational change induced by coordination to K^+^ (Figure 6, Cycle B). Indeed, a strong capture of K^+^ by the crown-ether ring during the catalytic cycle was confirmed as follows. When KCl (1 eq) was added to the selenoxide of **3** (**3^ox^**), which was generated by reacting **3** with H_2_O_2_ in CD_3_OD, complete spectral changes were observed in ^1^H and ^77^Se NMR, indicating the formation of the stable 3**^ox^**·K^+^ complex. This complex was easily converted to the **3**·K^+^ complex by the addition of excess DTT^red^ (see SI).

To rationalize the reason for the reduction in the redox catalytic activity of **3** in the presence of KCl, the density functional theory calculation was carried out at B3LYP/6-311+G(2df,p) with RCM (solvent = MeOH) for the diaxial and diequatorial conformers of DHS. As a result, it was found that the HOMO energy level of the diequatorial conformer (−6.15 eV) is lower than that of the diaxial conformer (−6.01 eV). Since the rate-determining step of the catalytic cycle of DHS is the oxidation process of DHS to DHS^ox^, the decrease in the catalytic activity can be reasonably explained by the conformational change in the DHS ring of **3** from diaxial to diequatorial due to the complexation of the 18-crown-6 ring to the metal ion.

## 3. Materials and Methods

### 3.1. General

^1^H (500 MHz), ^13^C (125.8 MHz) and ^77^Se (95.4 MHz) NMR spectra were recorded on a Bruker AV-500 spectrometer at 298 K in CDCl_3_ or CD_3_OD, using the solvent signals as internal standards. For ^77^Se NMR, dimethyl diselenide was used as an external standard. MALDI-TOF mass spectra were recorded on a JEOL JMS-S3000 mass spectrometer with a high-resolution mode. The sample was dispersed in the matrix of α-cyano-4-hydroxycinnamic acid (CHCA), with sliver nitrate as a cationization agent and polyethylene glycol as an internal standard. All reactions for the synthesis were monitored by thin-layer chromatography (TLC), which was performed on precoated sheets of silica gel 60 purchased from Merck Millipore. Gel permeation chromatography (GPC) was performed with a JAI LC-918 high-performance liquid chromatograph (HPLC) system equipped with JAIGEL-1H/2H preparative columns, using CHCl_3_ as an eluent at a flow rate of 3.5 mL/min. No calibration was applied. Racemic *trans*-3,4-dihydroxyselenolane (DHS) was synthesized according to the literature procedure [30,33]. Ditosylates, which were used for the preparation of DHS-crown-*n* (**1**–**4**), were synthesized according to the literature method, with slight modifications [34]. All other chemicals were used as purchased without further purification.

### 3.2. Synthesis of 10a,13a-trans-Decahydroselenopheno[3,4-b][1,4,7,10]tetraoxacyclododecine (***1***)

In a 30 mL round-bottom flask, DHS (52 mg, 0.31 mmol) and KOH (87 mg, 4 eq) were dissolved in *tert*-butyl alcohol (4 mL), and the mixture was stirred at 50 °C for 10 min. To the resulting solution, a solution of triethyleneglycol ditosylate (188 mg, 1.3 eq) in dichloromethane (1 mL) and then *tert*-butyl alcohol (6 mL) was added. The mixture was stirred at 80 °C for 1 h and then refluxed for 20 h. The resulting pale-yellow suspension was added with a saturated aqueous solution of ammonium chloride and extracted with AcOEt (×3). The combined organic layer was washed with brine and dried over magnesium sulfate. The crude product obtained was purified by GPC to afford DHS-crown-4 (**1**) as colorless crystals. Yield 35.4 mg (40%). m.p. 42–46 °C. Spectral for **1**: ^1^H NMR (CDCl_3_) δ 4.34 (m, 2H), 3.72 (m, 4H), 3.66 (m, 4H), 3.53 (m, 4H), 2.96 (dd, *J* = 5.0 and 10.0 Hz, 2H), 2.74 (dd, *J* = 5.0 and 10.0 Hz, 2H). ^13^C NMR (CDCl_3_) δ 86.2, 72.6, 70.5, 69.2, 24.1. ^77^Se NMR (CDCl_3_) δ 85.2. HRMS (MALDI-TOF-MS) *m*/*z* calcd for C_10_H_18_AgO_4_Se^+^ [M+Ag]^+^: 388.9416. Found: 388.9468. The molecular structure of **1** was determined by X-ray analysis for the crystals recrystallized from methanol (see below).

### 3.3. Synthesis of 13a,16a-trans-Dodecahydroselenopheno[3,4-b][1,4,7,10,13]pentaoxacyclopentadecine (***2***)

Following a similar procedure, DHS-crown-5 (**2**) was obtained from DHS (105 mg, 0.63 mmol) and tetraethyleneglycol ditosylate (320 mg, 1.0 eq) as a colorless oil. The crude product was purified by silica gel column chromatography (dichloromethane–methanol 9:1) and GPC. Yield 111 mg (55%). Spectral for **2**: ^1^H NMR (CDCl_3_) δ 4.39 (m, 2H), 3.81 (m, 2H), 3.73–3.65 (m, 14H), 3.06 (dd, *J* = 3.5 and 10.0 Hz, 2H), 2.91 (br d, *J* = 10.0 Hz, 2H). ^13^C NMR (CDCl_3_) δ 85.7, 71.4, 70.6, 70.5, 68.8, 25.1. ^77^Se NMR (CDCl_3_) δ 93.5. HRMS (MALDI-TOF-MS) *m*/*z* calcd for C_12_H_22_AgO_5_Se^+^ [M+Ag]^+^: 432.9678. Found: 432.9730.

### 3.4. Synthesis of 16a,19a-trans-Tetradecahydroselenopheno[3,4-b][1,4,7,10,13,16]hexaoxacyclooctadecine (***3***)

DHS-crown-6 (**3**) was obtained from DHS (112 mg, 0.67 mmol) and pentaethyleneglycol ditosylate (500 mg, 1.3 eq) as a colorless oil. The crude product was purified by silica gel column chromatography (dichloromethane–methanol 9:1) and GPC. Yield 141 mg (58%). Spectral for **3**: ^1^H NMR (CDCl_3_) δ 4.27 (m, 2H), 3.76–3.61 (m, 20H), 3.05 (dd, *J* = 4.0 and 10.0 Hz, 2H), 2.88 (br dd, *J* = 3.0 and 9.5 Hz, 2H). ^13^C NMR (CDCl_3_) δ 85.4, 71.1, 70.7, 68.9, 24.6. ^77^Se NMR (CDCl_3_) δ 90.4. HRMS (MALDI-TOF-MS) *m*/*z* calcd for C_14_H_26_AgO_6_Se^+^ [M+Ag]^+^: 476.9940. Found: 476.9982.

### 3.5. Synthesis of 19a,22a-Trans-hexadecahydroselenopheno[3,4-b][1,4,7,10,13,16,19]heptaoxacyclohenicosine (***4***)

DHS-crown-7 (**4**) was obtained from DHS (109 mg, 0.65 mmol) and hexaethyleneglycol ditosylate (531 mg, 1.3 eq) as a slightly yellow oil. The crude product was purified by silica gel column chromatography (dichloromethane–methanol 9:1) and GPC. Yield 158 mg (59%). Spectral for **4**: ^1^H NMR (CDCl_3_) δ 4.15 (m, 2H), 3.64–3.60 (m, 24H), 3.00 (dd, *J* = 4.0 and 10.0 Hz, 2H), 2.85 (dd, *J* = 4.0 and 10.0 Hz, 2H). ^13^C NMR (CDCl_3_) δ 85.4, 71.0, 70.9, 70.8, 70.8, 70.7, 69.1, 24.1. ^77^Se NMR (CDCl_3_) δ 88.3. HRMS (MALDI-TOF-MS) *m*/*z* calcd for C_16_H_30_AgO_7_Se^+^ [M+Ag]^+^: 521.0202. Found: 521.0330.

### 3.6. Complex Preparation

The prepared DHS-crowns (**1**–**4**) were separately dissolved in methanol. To each solution, an alkali metal salt (1 eq), i.e., KI or NaBr, was added, and the mixture was left still at room temperature to allow for the slow vaporization of the solvent. Crystals were grown and collected by filtration.

### 3.7. X-ray Analysis

Single-crystal X-ray diffraction data for DHS-crown-4 (**1**) and DHS-crown-4·0.5KI·H_2_O (**5**) were recorded on a Rayonix-MX225-HS CCD area detector coupled with a synchrotron radiation (*λ* = 0.70000 Å) and processed with HKL3000 [35] at 2D beamline in Pohang Accelerator Laboratory (PAL, Pohang, South Korea). Diffraction data for DHS-crown-4·0.5NaBr·2H_2_O (**6**), DHS-crown-5·0.5KI (**7**) and DHS-crown-6·KI (**8**) were measured with a Rigaku XtaLAB PRO P200 diffractometer using graphite monochromated Mo Kα radiation or multi-layer mirror monochromated Cu Kα radiation. The collected data were processed with CrysAlisPro (Rigaku Oxford Diffraction: Tokyo, Japan, 2015) or CrystalClear (Rigaku Corporation: Tokyo, Japan, 2015). The initial structures were solved by SHELXT (Version 2018/2) [36], except for DHS-crown-6·KI (**8**). The initial structure of DHS-crown-6·KI (**8**) was solved by SHELXS (Version 2013/1) [37]. The structure refinement was carried out by the full-matrix least-squares method on *F*^2^ using SHELXL (Version 2018/3) [37]. The CIF files for these compounds were deposited on CCDC (2248791–2248795).

### 3.8. ^1^H NMR Titration Study in CD_3_OD

DHS-crown-6 (**3**) (9.1 mg, 0.025 mmol) was dissolved in CD_3_OD (550 μL). The solution was transferred into an NMR test tube, and the ^1^H NMR spectrum was recorded. In a different small vial, an alkali metal salt, such as KI, CsCl or RbCl (0.10 mmol, 4 eq), was dissolved in CD_3_OD (200 μL). A small portion of D_2_O was added if the salt was imperfectly soluble. To make a solution of KBPh_4_, DMSO-d_6_ (120 μL) was used as the solvent. An aliquot of the solution was added to the solution of **3**, and the ^1^H NMR spectrum was recorded after standing for more than 30 min. The spectrum measurement was repeated with additions of the aliquots.

### 3.9. Redox Assay for DHS-crown-6 (**3**)

According to the literature [24], DTT^red^ (0.15 mmol) and a catalytic amount of **3** (0.015 mmol) were dissolved in CD_3_OD (1.1 mL) at 25 ºC in an NMR test tube. A portion of 30% H_2_O_2_ (0.15 mmol), the exact concentration of which was titrated in advance with potassium permanganate, was added to the test tube, and the 500 MHz ^1^H NMR spectrum was measured after certain periods of the reaction time. The well separated peaks of DTT^red^ and DTT^ox^ were integrated to calculate the ratio of the residual DTT^red^. In the assay for the KCl complex, KCl (0.015 mmol) was added with **3**. The reproductivity of the results was confirmed by repeating the experiments more than three times.

### 3.10. Theoretical Calculation

DFT calculation was performed using a Gaussian09 rev.B.01 program [38]. The geometry was fully optimized at the B3LYP/6-311+g(2df,p) level in methanol with a polarizable continuum model (PCM). The resulting structures for the two conformers of DHS were characterized as a stationary point with no imaginary vibrational frequency. A summary of the calculation results is provided in the Appendix A.

## 4. Conclusions

We synthesized a series of DHS-crown-*n* (*n* = 4–7) (**1**–**4**) and studied the complex formation with various alkali metal salts. According to the X-ray analysis, it was found that DHS-crowns change the conformation of the DHS ring from diaxial to diequatorial by coordination to an alkali metal ion. This conformational change was also confirmed by solution NMR study for DHS-crown-6 (**3**). In the literature, several crown ether derivatives, in which the crown-ether ring is fused by a five-membered ring system, such as tetrahydrofuran, pyrrolidine, and cyclopetane, have been reported [39,40,41]. However, the conformational transition of the five-membered ring system induced by the coordination to a guest ion was not reported previously, as per our literature survey. We further demonstrated the application of this interesting phenomenon to the control of the redox catalytic activity in the presence or absence of an alkali metal ion. Although completely turning off the activity by the addition of an alkali metal ion was not achieved here, this system will be valuable for further investigation for developing redox molecular switches or alkali metal ion sensors.

## Data Availability

The data presented in this study are available on request from the corresponding author.

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
