# Peer review of "Controlling the Redox Catalytic Activity of a Cyclic Selenide Fused to 18-Crown-6 by the Conformational Transition Induced by Coordination to an Alkali Metal Ion"

_molecules, 2023, doi:10.3390/molecules28083607_

Round 1

Reviewer 1 Report

The new DHS-crown-n (n = 4 to 7) were synthesized and characterized by NMR spectra, mass spectra and X-ray structure analysis. The complex formations of DHS-crown-6 with alkali metal salts were confirmed by NMR experiments and X-ray analysis. Furthermore, the redox catalytic activity of DHS-crown-6 was evaluated. The synthesis and characterization are doing well down.

Author Response

Thank you for your positive comments. We appreciate for your precious time to scrutinize the manuscript. Although no critical point was raised, we slightly modified the manuscript according to the comments from the other reviewers.

Reviewer 2 Report

The authors prepared  a series of DHS-crown-n compounds and explored the complex for- 301 mation with various alkali metal salts. The single crystal has been cultured and the conformational change was verified by NMR study. The results could be valuable for the development of redox molecular switches. Minor revisions should be considered before publication

1) As for compounds 2, 3 and 4, the authors should add the melting point for the molecules which would be important for following researchers. 

2) The detailed GPC purification condition needs to be given, such as  flow rate, calibration status etc. 

3) Since the redox activity of DHS is recorded, is it possible to perform cyclic voltammetry analysis in such sample ? It is known that oxidation and reduction peaks with high symmetry are observed in CV data and the well-reversible redox reaction has been achieved. 

Author Response

Thank you for positive and valuable comments toward publication of our paper.

Comment:

1) As for compounds 2, 3 and 4, the authors should add the melting point for the molecules which would be important for following researchers.

Our answer:

Compounds 2, 3 and 4 were obtained as a colorless or slightly yellow oil. This was described in the Materials and Method section. So, the melting points could not be measured.

Comment:

2) The detailed GPC purification condition needs to be given, such as  flow rate, calibration status etc.

Our answer:

Thanks for this comment. We added the information in the Materials and Method section as follows.

“Gel permeation chromatography (GPC) was performed with a JAI LC-918 high-performance liquid chromatograph (HPLC) system equipped with JAIGEL-1H/2H preparative columns, using CHCl3 as an eluent at a flow rate of 3.5 mL/min. No calibration was applied.”

Comment:

3) Since the redox activity of DHS is recorded, is it possible to perform cyclic voltammetry analysis in such sample ? It is known that oxidation and reduction peaks with high symmetry are observed in CV data and the well-reversible redox reaction has been achieved.

Our answer:

It would be very interesting to measure the CV voltammograms for DHS-crowns in the presence or absence of an alkali metal ion. However, we do not have the device for CV measurement unfortunately. So, we would like to consider this in future with a collaboration with other research groups.

Reviewer 3 Report

The 3,4-dihydroxyselenolan-crown family was synthesized and complex formation with various alkali metal salts was studied. It was found that DHS-crowns change the conformation of the DHS ring from diaxial to diequatorial by coordination with an alkali metal ion. This conformational change was previously not observed and contributed well to the novelty of the article. This is a well-written and well-designed article with new facts and new interesting macrocyclic compounds with glutathione peroxidase-like activity. In my opinion, the article will be very interesting for readers of Molecules and can be accepted in the present form. 

Author Response

Thank you for warm comments. Yes, I hope that this paper will stimulate the readers of Molecules. Although no revision was requested, we slightly modified the manuscript according to the comments from the other reviewers.